# Speech Perception Changes in the Acoustically Aided, Nonimplanted Ear after Cochlear Implantation: A Multicenter Study

**DOI:** 10.3390/jcm9061758

**Published:** 2020-06-05

**Authors:** Mario A. Svirsky, Arlene C. Neuman, Jonathan D. Neukam, Annette Lavender, Margaret K. Miller, Ksenia A. Aaron, Piotr H. Skarzynski, Katarzyna B. Cywka, Henryk Skarzynski, Eric Truy, Fabien Seldran, Ruben Hermann, Paul Govaerts, Geert De Ceulaer, Francois Bergeron, Matthieu Hotton, Michelle Moran, Richard C. Dowell, Maria Valeria Schmidt Goffi-Gomez, Ana Tereza de Matos Magalhães, Rosamaria Santarelli, Pietro Scimemi

**Affiliations:** 1Department of Otolaryngology-Head and Neck Surgery, NYU Grossman School of Medicine, New York University, New York, NY 10016, USA; arlene.neuman@gmail.com (A.C.N.); jonathan.neukam@nyulangone.org (J.D.N.); 2Neuroscience Institute, NYU Grossman School of Medicine, New York University, New York, NY 10016, USA; 3Center for Neural Science, New York University, New York, NY 10003, USA; 4Cochlear Americas, Denver, CO 80124, USA; annettezeman@gmail.com; 5Human Auditory Development Lab, Boys Town National Research Hospital, Omaha, NE 68131, USA; margaret.miller@boystown.org; 6Otolaryngology Head and Neck Surgery, Stanford Medicine, Stanford, CA 94305, USA; ksenia.prosolovich@gmail.com; 7Department of Teleaudiology and Screening, World Hearing Center, Institute of Physiology and Pathology of Hearing, Warsaw/Kajetany, 02-042 Warsaw, Poland; p.skarzynski@ifps.org.pl; 8Heart Failure and Cardiac Rehabilitation Department, Medical University of Warsaw, 02-091 Warsaw, Poland; 9Institute of Sensory Organs, Kajetany, 05-830 Warsaw, Poland; 10Department of Otorhinolaryngosurgery, World Hearing Center, Institute of Physiology and Pathology of Hearing, Warsaw/Kajetany, 02-042 Warsaw, Poland; k.cywka@ifps.org.pl (K.B.C.); h.skarzynski@ifps.org.pl (H.S.); 11INSERM U1028, Lyon Neuroscience Research Center, Equipe IMPACT, 69000 Lyon, France; eric.truy@chu-lyon.fr (E.T.); rubenhermann@gmail.com (R.H.); 12CNRS UMR5292, Lyon Neuroscience Research Center, Equipe IMPACT, 69000 Lyon, France; 13Medel France, 06160 Antibes, France; fabien.Seldran@medel.com; 14De Oorgroep, Herentalsebaan 75, B-2100 Antwerp-Deurne, Belgium; dr.govaerts@eargroup.net (P.G.); deceulaer@eargroup.net (G.D.C.); 15Université Laval, 1050, Avenue de la Médecine, Québec, QC G1V 0A6, Canada; francois.bergeron@rea.ulaval.ca (F.B.); mathieu.hotton.1@ulaval.ca (M.H.); 16The University of Melbourne, Melbourne, VIC 3053, Australia; mmoran@unimelb.edu.au (M.M.); rcd@unimelb.edu.au (R.C.D.); 17Royal Victorian Eye & Ear Hospital, East Melbourne, VIC 3002, Australia; 18The HEARing Co-Operative Research Centre, Melbourne, VIC 3053, Australia; 19Hospital das Clinicas da Faculdade de Medicina da Universidade de São Paulo, Brazil. Av. Dr. Eneas de Carvalho Aguiar, 255, 05403-000, São Paulo, Brazil; goffigomez@uol.com.br (M.V.S.G.-G.); anatereza32@gmail.com (A.T.d.M.M.); 20Department of Neurosciences, School of Medicine and Surgery, University of Padua, 35128 Padua, Italy; rosamaria.santarelli@gmail.com (R.S.); pietro.scimemi@unipd.it (P.S.); 21Otorhinolaryngology and Audiology Unit, “Santi Giovanni e Paolo” Hospital, 30126 Venice, Italy

**Keywords:** cochlear implants, hearing aid, bimodal, speech perception, auditory neglect, deafness, multicenter

## Abstract

In recent years there has been an increasing percentage of cochlear implant (CI) users who have usable residual hearing in the contralateral, nonimplanted ear, typically aided by acoustic amplification. This raises the issue of the extent to which the signal presented through the cochlear implant may influence how listeners process information in the acoustically stimulated ear. This multicenter retrospective study examined pre- to postoperative changes in speech perception in the nonimplanted ear, the implanted ear, and both together. Results in the latter two conditions showed the expected increases, but speech perception in the nonimplanted ear showed a modest yet meaningful decrease that could not be completely explained by changes in unaided thresholds, hearing aid malfunction, or several other demographic variables. Decreases in speech perception in the nonimplanted ear were more likely in individuals who had better levels of speech perception in the implanted ear, and in those who had better speech perception in the implanted than in the nonimplanted ear. This raises the possibility that, in some cases, bimodal listeners may rely on the higher quality signal provided by the implant and may disregard or even neglect the input provided by the nonimplanted ear.

## 1. Introduction

Indications for cochlear implantation have changed significantly since these devices were introduced to clinical practice. Initially, only patients with profound to total hearing loss were implanted, but as outcomes of cochlear implantation improved with the development of newer devices, electrodes, and stimulation strategies, it became more common for patients with residual hearing to receive a cochlear implant. In 2014, the FDA expanded cochlear implantation criteria and gave premarket approval for implantation in ears that have useable acoustic hearing in the low frequencies [1]. Most recently, FDA approval was granted for cochlear implantation for single-sided deaf patients, defined as those who have “profound sensorineural hearing loss in one ear and normal hearing or mild sensorineural hearing loss in the other ear” [2]. The new audiometric criteria for cochlear implantation have yielded a quickly expanding group of persons with cochlear implants and most of them are “bimodal patients”: those who have usable acoustic hearing in the contralateral ear, typically aided by a hearing aid. As these criteria continue to evolve, it is likely that an even larger proportion of future CI patients will have usable hearing in the contralateral ear. The public health significance of this new population is evident when we compare the numbers: more than 800,000 Americans report being unable to hear normal conversation even while using a HA, in addition to the 180,000 classified as “deaf” [3]. In contrast, only 58,000 adults were reported to have received CIs in the United States as of 2012 [4].

Initial research on bimodal hearing indicated that even persons with severe to profound hearing loss in the nonimplanted ear could obtain bimodal benefit (e.g., [5,6,7,8,9,10,11,12,13,14]). More recent research has focused on bimodal benefit obtained by persons with a unilateral cochlear implant who have mild to severe hearing loss in the nonimplanted ear. In these cases, bimodal benefit is obtained from amplification to the low frequency region, even when speech recognition performance with the HA alone is very poor (e.g., [15,16,17,18,19,20,21]). Bimodal benefit is also observed in patients with asymmetric hearing loss who receive a cochlear implant in their worse ear [18,19,22,23,24,25,26,27].

Bimodal benefit on speech recognition tasks is characterized by a great deal of variability among individuals. For example, in a report by Gifford et al. (2010) [28] 14 of 18 bimodal patients showed higher Consonant–Nucleus–Consonant (CNC) word scores in the bimodal condition than in the CI-only condition. Benefit ranged from 6% to 45%. The mean improvement for the group was 15%. Dorman et al. (2015) [29] reported results of CNC tests from 105 bimodal patients with varying amounts of acoustic hearing in the nonimplanted ear. Mean bimodal benefit on the CNC test material was related to the amount of low frequency acoustic hearing (125, 250 and 500 Hz). Those with the best low-frequency acoustic hearing obtained a mean benefit of 16%, while those with mean low frequency thresholds exceeding 61 dB hearing level (HL) obtained a mean benefit of 4%. But benefit of individual participants ranged from −15% to 55%. In general, the benefit was greater for speech recognition in noise. For example, on the AZBio test material (Auditory Potential, LLC, Goodyear, AZ 85338) presented at +10 dB signal-to-noise ratio (SNR) [10,16,17] mean benefit was 26% for those with the highest level of acoustic hearing, but only 8% for those with the least low-frequency acoustic hearing. Even though many derive some benefit from the simultaneous use of a CI and a contralateral HA, for some, use of the HA results in decreased performance (e.g., [12,30,31,32,33]). Taken together, these results suggest that many patients obtain levels of bimodal benefit that are clinically relevant whereas others simply use “best ear listening” (with the best ear being the one with the cochlear implant, typically) or may even experience interference from the weaker ear signal (usually the HA ear).

Some of these speech perception results may be driven by asymmetric hearing, defined as a situation where one ear receives (and conveys to the brain) much more information than the other one. Auditory asymmetries are known to have consequences that have been demonstrated in animal models and human listeners [21,34,35]. This can happen when the input asymmetry is due to actual physiological asymmetries, as in the examples just referenced, or even when it is imposed by unilateral acoustic aiding in the face of bilateral, symmetric hearing impairment. The latter phenomenon has been called “late onset auditory deprivation”, and it was first reported by Silman et al. (1984) [36] (for a review, see Neuman, 1986 [37]). The study observed that many monaurally aided adults with bilateral symmetric sensorineural hearing impairment exhibited decreases in speech-recognition ability in the unaided ear, but not in the aided ear. This happened even though the unaided thresholds remained similar in the aided and the unaided ear.

We first reported in 2011 [38] an interesting pattern in bimodal cochlear implant users that was reminiscent of Silman et al.’s (1984) results. A retrospective study of bimodal CI users found statistically significant speech perception decreases in the nonimplanted ear in 11 out of 32 patients after cochlear implantation. It is notable that this analysis excluded any patients whose unaided thresholds in the nonimplanted ear changed by more than 10 dB after implantation, suggesting the possibility that those changes in speech perception could not be completely explained by changes in hearing sensitivity. These results prompted the organization of the multicenter study we report here, aimed at tracking possible changes in speech perception in the acoustically aided ear of bimodal patients after cochlear implantation. Importantly, this study only includes data from patients where there was some indication of correct hearing aid functioning. The main goals were: first, to determine whether the initial observation would be replicated in a larger, multicenter sample, and second, to start teasing out possible explanations for the phenomenon.

## 2. Experimental Section

### 2.1. Inclusion Criteria

Participants were adults with post-lingual hearing loss who used a cochlear implant (CI) in one ear and a hearing aid (HA) in the contralateral ear (bimodal users). To be included in the study, documentation of hearing pre- and post-implant surgery was required for the implanted ear and the non-implanted ear. These longitudinal measures included documentation of unaided pure tone thresholds in the nonimplanted ear, as well as speech recognition performance pre- and post-implantation in the CI ear, the HA ear, and in the bimodal condition. An additional requirement was documentation that showed the hearing aid was functioning properly during testing. This requirement could be met in one of four possible ways: 1) in-the ear measures showing that hearing aid gain was adequate; 2) aided and unaided thresholds in the HA ear; 3) unaided speech perception scores in the HA ear pre- and post-implant (if a drop in speech perception scores was not accompanied by a similar drop in unaided scores in the same ear, we assumed that this was due to hearing aid malfunction or improper fitting and the data were excluded); or 4) indication from an audiologist or hearing aid technician that the HA was operating correctly during the evaluation.

### 2.2. Subjects

There were 132 subjects: 68 male and 64 female. The average age at implantation was 56.8 years (minimum 15, maximum 89). Table 1 and Table 2 show how the 132 subjects are classified by etiology, device, electrode, and speech processor. The number of post-implant sessions ranged from 1 to 11, with an average of 2.5 sessions. The average duration of post-implant follow-up for the hearing aid ear was 3.09 years, ranging from just one month to as much as 11 years. Figure 1 shows the average preoperative and postoperative unaided audiograms in the nonimplanted ear. The difference in pure-tone average is 2.5 dB. Given the average age of the subjects, the likelihood that some of them had progressive hearing loss, and the fact that some hearing loss is expected due to presbycusis (as subjects were followed for over three years, on average), this 2.5 dB difference does not seem unusual. Verification of hearing aid status was done by comparing aided and unaided thresholds for 81 subjects and by audiologist report for 41 subjects. In-the-ear verification or comparison of aided and unaided speech scores was not used for any of the subjects. Lastly, it was not possible to obtain information about hearing aid status for ten of the subjects, so their data were excluded from several of the analyses, as reported below.

### 2.3. Speech Perception Tests Used at Each Site

Data was collected at eight different testing sites. Speech testing consisted of words or sentences in quiet presented via audition alone (no visual cues) at 65 dB HL (unless otherwise noted) and zero degrees azimuth. See Appendix A
Table A1 for a summary of speech testing materials and sample size at each site.

### 2.4. Pre- vs. Post-Implantation Comparisons

Speech perception scores before implantation were compared to post-implantation speech scores. This was done separately for three conditions: implanted ear only (in the case of pre-implantation data, this means the ear to be implanted, using acoustic amplification when appropriate), nonimplanted ear only, and bilateral. These comparisons were made in two different ways: comparing pre-implant to post-implant scores averaged over all postoperative sessions, or just to the latest available postoperative score. These two ways of comparing preoperative to postoperative data yielded similar results. The statistical significance of pre- to postoperative differences was assessed at the group level using paired t-tests, or the Wilcoxon signed rank test when data failed to follow a normal distribution. Follow-up tests were conducted on subsets of the data to help tease out possible reasons for preoperative to postoperative changes in speech perception in the nonimplanted ear. Specifically, comparisons were made for the original dataset of 132 subjects who had pre- and postoperative scores in the nonimplanted ear and for the same group after excluding any evaluation session when correct hearing aid operation could not be ascertained (N = 122). By design, all datapoints obtained postoperatively in the nonimplanted ear were included if the pure-tone average (PTA) did not differ from the preoperative PTA by more than 25 dB. To better rule out the effect of progressive hearing loss, analyses of speech scores in the nonimplanted ear were conducted after removing datapoints showing an increase of more than 20 dB PTA. The remaining N was 120 for the preoperative to average postoperative comparison, and N = 118 for the preoperative to latest postoperative comparison. The process of elimination was repeated for datapoints showing a PTA increase of more than 10 dB, and the resulting numbers of the remaining subjects were N = 107 and N = 104, when comparing preoperative scores to the average postoperative and latest postoperative scores, respectively. Finally, this was also done after excluding datapoints showing a change of more than 5 dB in PTA. The resulting numbers of subjects in this case were N = 92 and N = 85 for the two types of comparison.

Pre- to postoperative comparisons were also conducted for each subject and condition (implanted ear, nonimplanted ear, bilateral) by generating critical difference tables as described by Carney and Schlauch [39]. The critical difference tables were generated using a Matlab program that works in the general case when different numbers of items are used in the two scores to be compared, which can happen, for example, when comparing a score obtained with one 50-word list to a score obtained as the average of three 50-word lists (the Matlab code can be downloaded from https://osf.io/fha47/?view_only=3d5c2e239a64496481ea657215913469). This was done separately for each test. To determine whether two scores are significantly different, computer simulation was used to generate confidence intervals based on the number of observations used to obtain each score. Let n1 and n2 represent the number of observations from which those two scores were obtained, respectively. Next, consider all possible scores (as proportions) that can be obtained for these numbers of observations. For example, if n1 = 50, then the possible scores are 0, 0.02, 0.04, …, etc., and if n2 = 25, then the possible scores are 0, 0.04, 0.08, …, etc. Let *p1*(i) represent the set of possible scores for n1 where i = 1, 2, …, n1 + 1, and *p2*(j) represent the set of possible scores for n2 where j = 1, 2, …, n2 + 1. For every pair of possible scores from *p1*(i) and *p2*(j), one calculates a mean absolute difference score using computer-generated binomially distributed pseudorandom numbers for *p1*(i) and *p2*(j), repeated over many iterations. That is, on one iteration, two binomially distributed numbers are generated, one based on *p1*(i) and n1 and the other based on *p2*(j) and n2. The numbers are then converted to proportions by dividing by the respective number of observations, n1 and n2. The absolute value of the difference between these two scores is then recorded. This process is repeated for any number of desired iterations, e.g., 40,000. For this set of 40,000 absolute difference scores, one can calculate a z-score equal to the mean absolute difference divided by the standard deviation estimate across absolute difference scores. Having generated a z-score for each *p1*(i) and *p2*(j), one can find the values of *p1*(i) and *p2*(j) that produce z-scores exceeding a critical value (e.g., 1.96 for a normal distribution and a two-tailed test using a *p*-value of 0.05). For p1(i), the values of *p2*(j), whose z-scores fall within +1.96 and −1.96, define the critical range.

Some data from two of the sites were obtained using scores of correct keywords in sentences. The TAM test has between 114 and 125 scored words per list and the CUNY sentence test has 102 words per list, on average. In those cases, critical significance tables are used N = 40, which have been shown to be an appropriately conservative assumption for a sentence test like AzBio (which has 142 words per list) when scoring it using the percentage of correct keywords [40].

The change in speech scores in the nonimplanted ear (which was typically a decrease) was correlated with three variables of interest: the preoperative to postoperative change in pure-tone average, the speech score in the implanted ear, and the implanted ear advantage (difference in speech score between the implanted and the nonimplanted ear). The latter correlation was also calculated after partialling out the effect of the initial score in the nonimplanted ear. It was expected that the pattern of correlations might help tease out the effect of some possible causes of speech perception drops in the nonimplanted ear. A significant correlation between drops in speech scores and in pure-tone average in the nonimplanted ear would suggest that the progression of hearing loss might be at least partly responsible for decreases in speech scores. Conversely, a significant correlation between drops in speech scores in the nonimplanted ear and either speech scores in the implanted ear or the implanted ear advantage would suggest that the speech perception drops in question are at least partly due to a more central mechanism: namely, neglect of the poorer ear.

Lastly, and for the sake of completeness, the change in speech scores in the nonimplanted ear was also correlated with several preoperative demographic variables: age at cochlear implantation, age at moderate hearing loss, age at profound hearing loss, preoperative aided and unaided PTA in both the implanted and the nonimplanted ear, whether cochlear implantation was done using a cochleostomy or a round window approach, whether the left or the right ear was implanted, whether a hearing was used in the ear to be implanted, years of education, and gender.

This study was approved by the New York University Medical Center Institutional Review Board (S12-02951).

## 3. Results

Figure 2 shows a comparison of preoperative vs. postoperative speech perception scores. The top panels compare preoperative scores to the average of all postoperative scores for a given subject. The bottom panels compare preoperative scores to the latest available postoperative score. As we will see, the conclusions are unaffected whether we examine scatterplots in the top or the bottom panel. Sets of panels, left to right, indicate speech perception scores obtained with the cochlear implant ear, both ears at the same time, or the acoustically stimulated ear. Each symbol represents one subject and color is used to indicate whether the postoperative scores are significantly different (*p* < 0.05) from preoperative scores for each subject. Green indicates that the postoperative score was significantly higher, red indicates that the postoperative score was significantly lower, and blue indicates no significant difference between preoperative and postoperative scores.

The left panels, not surprisingly, show that speech perception in the implanted ear improved significantly for the vast majority of subjects (94 out of 102, regardless of whether we examine average postoperative performance or the latest available score). This expected result is more impressive when we consider that an additional 29 subjects could not be included because their preoperative score was listed as “could not test”, which almost always means that the subject in question did not have sufficient residual hearing in the ear to be implanted to understand any words without lipreading, so the score would have been zero. The same happened with speech perception in the binaural condition shown in the center panels: 62 out of 82 subjects showed statistically significant improvement when comparing to the latest datapoint, and this was the case for 61 subjects when comparing against the average postoperative score. For many of the remaining 19 subjects, it would have been impossible or at least very difficult to show a statistically significant improvement due to ceiling effects. Again, 41 subjects were not evaluated preoperatively in the binaural condition.

A very different picture is seen when examining preoperative to postoperative changes in the nonimplanted ear (right panels). Here, we see that 41 subjects scored significantly lower (*p* < 0.05) in their latest postoperative evaluation than they did preoperatively (right bottom panel) and 38 subjects did so when comparing their preoperative score to their average postoperative score. Twelve subjects obtained significantly higher scores in their latest postoperative evaluation compared to the preoperative one, and 11 showed a similar change when comparing their preoperative score to their average postoperative score. Remember that some of these “statistically significant” differences would have been observed even in the absence of any real change. Because 132 subjects were evaluated and a p-value of 0.05 was used as a threshold, we would expect an average of three subjects (2.5% of 132, rounded up to the nearest integer) to show significant increases and another three to show significant decreases. Thus, the number of subjects with postoperatively decreased scores in the hearing aid ear far exceeds what would be expected due to random variation in the absence of a real effect. This was true both at the site with the highest number of subjects (NYU, where 18 out of 53 subjects had significantly lower scores in the latest postoperative evaluation than in the preoperative evaluation) and in the rest of the sites, where this was true for 23 out of 79 subjects.

All preoperative-to-postoperative group differences were highly significant (*p* < 0.001). Postoperative scores were significantly higher for the implanted ear and for the binaural condition, and lower for the nonimplanted ear. Because the results in the nonimplanted ear were somewhat unexpected, we also examined the preoperative–postoperative differences using increasingly strict criteria. First, we excluded ten subjects for whom proper hearing aid operation could not be confirmed. Then, we excluded subjects who showed more than 20, 10, or 5 dB of hearing loss in the nonimplanted ear during the course of the study. Figure 3 and Table 3 show the results of these additional analyses. As can be observed, after cochlear implantation there is a robust decrease in speech perception in the nonimplanted ear. The effect is highly significant whether we consider the latest available point for each subject or the average of all postoperative scores, and whether we include or exclude data from subjects whose hearing loss became more pronounced during the study. We do observe that this group effect is somewhat less pronounced (but still present) when we only include data from subjects whose PTA did not change by more than 5 dB.

To help tease out some of the reasons for the decrease in speech perception in the nonimplanted ear, correlations between that decrease and two relevant variables were examined. For this analysis we only included datapoints when there was evidence that the hearing aid was functioning properly. The left panel of Figure 4 shows the decrease in speech perception in the nonimplanted ear speech score decrease as a function of change in pure-tone average for the same ear. As the legend indicates, there was a correlation in the expected direction (increase in PTA is associated with a drop in speech perception score in the nonimplanted ear) but the correlation is weak (r = +0.157) and fails to reach statistical significance (*p* = 0.08). The middle panel shows the speech perception drop in the nonimplanted ear as a function of the implanted ear advantage (that is, the extent to which speech perception is better in the implanted than in the nonimplanted ear). In other words, we examine the extent to which asymmetry between ears (with the implanted ear providing more information than the nonimplanted ear) may be associated with speech perception drops in the nonimplanted ear. That association seems to be supported by the correlation in the middle panel (r = +0.507, *p* < 0.001), but that may be a partially spurious effect driven by the fact that both variables are related to postoperative hearing aid scores. We address this issue in two ways. First, we found that the correlation becomes weaker but remains significant when partialling out the effect of postoperative hearing aid scores (r = +0.247, *p* = 0.0067). Second, there was also a positive correlation between average postoperative CI speech scores and the speech perception drop in the nonimplanted ear (r = +0.217, *p* = 0.0174; the corresponding scatterplot is shown in the right panel of Figure 4).

None of the other preoperative demographic variables were significantly correlated with the drop in speech perception in the nonimplanted ear, although one of them came close. The drop in speech perception between the preoperative evaluation and the average postoperative score showed a weak (r = +0.106) nonsignificant correlation (*p* = 0.059, NS) with age at cochlear implantation.

## 4. Discussion

The most interesting result we have presented is in the right panels of Figure 2 and Figure 4. Average speech perception scores in the nonimplanted ear showed an unexpected and statistically significant decrease after cochlear implantation. In principle, we do not expect any changes in the nonimplanted ear, particularly in the absence of any major changes in hearing sensitivity in that ear.

Results for the other two conditions (CI-only and bimodal) gave the expected pre- vs. postoperative improvements in speech perception scores (left and center panels of Figure 2), whether we consider the average postoperative score for each individual (top row of panels in Figure 2) or the latest datapoint (bottom row of panels). The vast majority of the subjects showed improvements that were both statistically significant and meaningful in a practical sense.

There are several potential factors that may underlie these unexpected changes in speech perception in quiet in the nonimplanted ear. Determining the precise combination will likely require a prospective follow-up study, but we can speculate about various possibilities. One of them is hearing aid malfunction, or at least improper hearing aid fitting. Recall, however, that the present study attempted to minimize this factor by requiring documentation that the hearing aid was functioning properly during testing, and that this requirement was met by each datapoint that was included in the study. This makes it less likely that the drops in speech perception were due to hearing aid malfunction or improper fitting. One caveat, however, is that many of the datapoints were confirmed simply by an audiologist’s indication that the hearing aid was functioning properly. A prospective study should require more reliable confirmation of proper hearing aid function, ideally by using in-the-ear measures showing that the fitting targets have been achieved.

A second possibility is that of progressive hearing loss in the nonimplanted ear, which could conceivably result in speech perception drops even with a properly fit hearing aid. However, it seems very unlikely that this factor could explain all of the observed change in speech perception in the nonimplanted ear, for three reasons. First, Figure 1 shows that the average group change in pure tone average in the nonimplanted ear was quite small: only 2.5 dB. Second, the scatterplot in the left panel of Figure 4 shows that very few individuals experienced pure-tone average drops of 10 dB or more, and even those who did were not more likely to also experience decreases in speech perception than those who did not. The correlation between change in pure-tone average and speech perception in the nonimplanted ear failed to reach statistical significance. Third, Figure 3 shows that gradually tightening the requirements for data inclusion did not substantially change the main outcome. We first excluded 10 patients for whom hearing aid functionality was not confirmed, and this left the drop in hearing aid speech scores largely unaffected (for example, the drop from preoperative to latest postoperative data point only changed by a small fraction of a percentage point). It is interesting to note that as we then remove patients whose PTA increased by more than 20, 10, or even 5 dB over time after cochlear implantation, the group average drop in speech perception in the nonimplanted ear goes from 9.9% to 8.6%, 8.0%, and 6.7% (bottom panel of Figure 3). Two observations relate to this result. First, even though drops in speech perception in the nonimplanted ear were not significantly correlated with increases in PTA, it seems quite possible that hearing loss progression underlies a fraction of the observed result. The second observation is that hearing loss progression is unlikely to explain the totality of the observed effect. There must be other mechanisms at play.

A third possibility is that this phenomenon might be at least conceptually related to that of late onset auditory deprivation, first reported by Silman et al. (1984) [36]. This refers to decreases in speech perception after a prolonged lack of amplification in the unaided ears of some unilaterally aided patients with bilateral hearing loss. It is at least possible that some of the patients in the present study stopped using their hearing aids or started using them less consistently after implantation. This may have happened after speech perception with the cochlear implant became much better than with their acoustically stimulated ear.

Alternatively (and this is a fourth possibility), it may be that speech perception decreases happened despite the continued use of acoustic input in the nonimplanted ear. One possible mechanism has to do with the adaptation that is required from postlingually deafened cochlear implant patients who have acoustic hearing in the nonimplanted ear. The frequency-place function imposed by a cochlear implant is typically different from the normal hearing frequency-place [41], and it is possible that some listeners may have difficulty adapting to different functions in each ear, the one with the cochlear implant and the one with acoustic amplification. In their case, the adaptation process that allows them to understand speech better with the cochlear implant may be maladaptive for the purpose of processing acoustic input.

The finding that drops in speech perception in the nonimplanted ear were correlated with implanted ear advantage (that is, the extent to which perception in quiet is better in the implanted ear than in the nonimplanted, acoustically aided ear) or even with the raw speech perception scores in the implanted ear suggests that the third or fourth possibilities discussed above (and which could be described as bimodal auditory neglect) may underlie some of the observed changes. In other words, to the extent that speech perception in the implanted ear may be much better than in the acoustically aided ear, this may result in a central auditory process wherein the brain relies on the higher quality signal provided by the implant and disregards the input provided by the nonimplanted ear.

A number of important caveats and suggestions for future research must be listed. Ideally, future studies should obtain more precise determinations of hearing aid fitting, particularly the gold standard of in-the-ear verification. Additionally, and given that both modern cochlear implants and hearing aids can log data about device use, it would be good to determine the extent to which the observed speech perception drops in the nonimplanted ear may or may not be related to the daily duration of hearing aid use, or the difference in daily duration of cochlear implant and hearing aid use.

Taken together, the present results suggest that there are considerable individual differences in the way acoustic and electrical auditory input interact. Better understanding of this interaction may help guide future clinical practice in cochlear implant patients who have usable residual hearing.

## Figures and Tables

**Figure 1 jcm-09-01758-f001:**
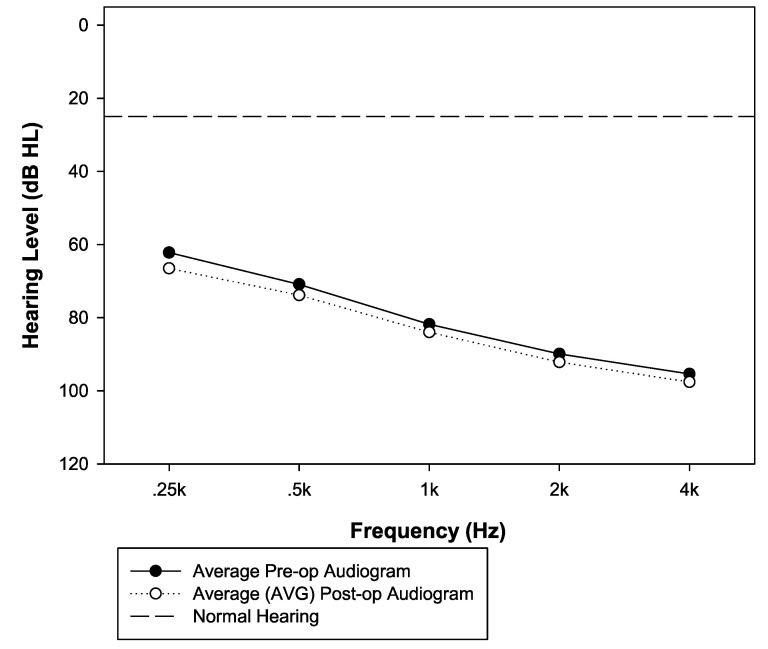
Group mean of preoperative and average postoperative unaided thresholds.

**Figure 2 jcm-09-01758-f002:**
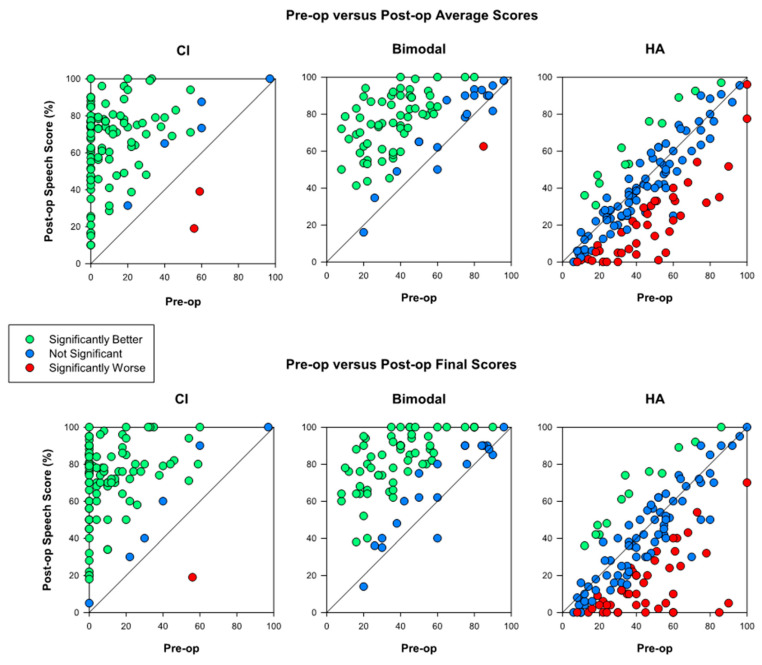
Postoperative vs. preoperative speech perception scores with the implanted ear only (left), both ears (bimodal condition, center), and the nonimplanted ear only (right). The top panels present average postoperative scores in the y-axis whereas the bottom panels show the latest available postoperative datapoint.

**Figure 3 jcm-09-01758-f003:**
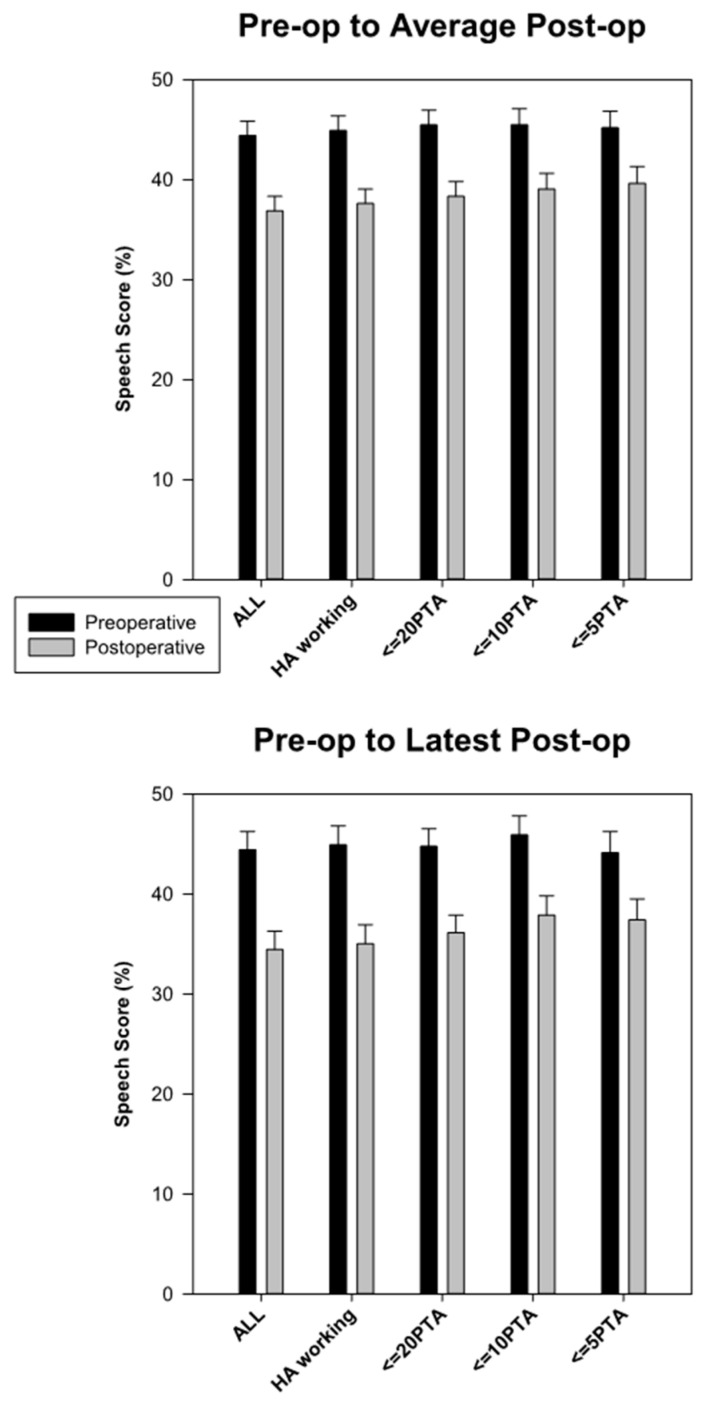
Postoperative speech perception scores in the nonimplanted ear are lower than preoperative scores even after excluding subjects whose hearing aid functioning could not be verified, or after excluding datapoints from subjects who showed decreases in pure-tone average greater than 20, 20, or 5 dB. The top panel compares preoperative data to average postoperative data and the bottom panel compares it to the latest available postoperative data.

**Figure 4 jcm-09-01758-f004:**
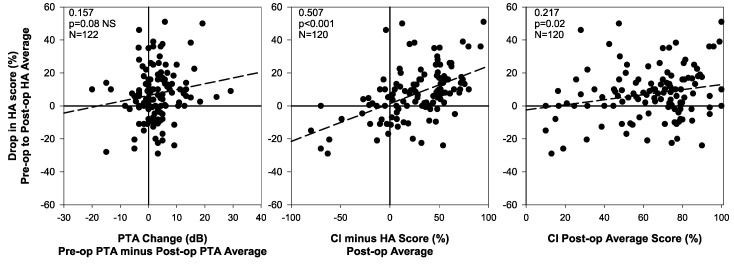
Drop in postoperative speech perception scores in the nonimplanted ear as a function of: change in pure-tone average (left panel), implanted ear advantage (or degree to which speech perception is better in the implanted than in the nonimplanted ear, center panel), and speech perception score in the implanted ear (right panel).

**Table 1 jcm-09-01758-t001:** Etiology.

Etiology	
Unknown	52
Genetic	19
Noise	14
Otosclerosis	13
Meniere’s	4
Meningitis	4
Progressive	4
SSNHL	4
Chronic Otitis Media	3
Head Trauma	3
Ototoxicity	3
Other	9
Total	132

**Table 2 jcm-09-01758-t002:** Cochlear implant brand, electrode, and speech processor.

	Electrode		Processor	
**AB**	1j	19	Harmony	12
	other	2	other	9
**Med-El**		25	Opus 2	21
			other	4
**Cochlear**	Contour/Contour Advance	63	Freedom	37
	other	13	N5	23
			other	16
**Neurelec**		10	Saphyr	10
**TOTAL**		132		132

**Table 3 jcm-09-01758-t003:** Preoperative and postoperative speech perception scores in the nonimplanted ear (% correct).

	AVERAGE					LATEST				
	N	Pre-op	Post-op	Difference	*p*-Value	N	Pre-op	Post-op	Difference	*p*-Value
**All subjects**	132	44.42	36.89	−7.53	<0.001	132	44.42	34.46	−9.96	<0.001
**HA working**	122	44.92	37.62	−7.30	<0.001	122	44.92	35.03	−9.89	<0.001
**< = 20 PTA**	120	45.48	38.34	−7.15	<0.001	118	44.77	36.14	−8.63	<0.001
**< = 10 PTA**	107	45.51	39.08	−6.44	<0.001	104	45.90	37.91	−7.99	<0.001
**< = 5 PTA**	91	45.20	39.65	−5.55	0.00129	85	44.13	37.41	−6.72	0.00195

Left columns show comparison to average postoperative scores, right columns show comparison to latest available score. Rows show comparisons for progressively restricted subsets of subjects.

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
