# Peer review of "Speech Perception Changes in the Acoustically Aided, Nonimplanted Ear after Cochlear Implantation: A Multicenter Study"

_jcm, 2020, doi:10.3390/jcm9061758_

Round 1

Reviewer 1 Report

General commants:
The manuscript reports on a multicenter study of speech recognition in listeners who use a cochlear implant (CI) in one ear and a hearing aid (HA) in the other. Of interest is the suggestion from previous studies, including one by some of the authors of the present manuscript, that speech recognition after implantation may decline in the HA ear. That suggestion was confirmed, at least for some listeners. The data were carefully analyzed, and the results were straightforward and clearly presented. The discussion provides a thoughtful analysis and interpretation of the results. The manuscript provides important information on a timely topic.

I think the manuscript would be strengthened by including some information that compares test sites and/or test materials (sentences vs words, etc). I realize that several sites had small numbers of participants, but at a minimum it might be interesting to compare the NYU data to everything else. Are the results consistent across settings, or biased by the largest subset of the data? A table presenting that information might also include a concise summary of the text in lines 152-183.

Specific comments:

line 60: typo, "2019implantation"?

line 180-181: were any of the NYU patients included in the 2011 preliminary report (ref 38)? Either way, that would be useful information for the reader.

line 209 awkward sentence, could probably just delete "Statistical significance of"

line 234-235: extraneous highlighted text?

Reviewer 2 Report

Speech perception changes in the acoustically aided, non-implanted ear after cochlear implantation: a multicentre study.

The paper describes a multicentre study over several years to observe the speech perception of the contralateral non implanted ear of cochlear implant patients. Post-operative changes in speech perception were measured in several variations, crucially in the contralateral ear. Results show a (surprising) decrease in speech perception ability in the non-implanted ear.

This is a well written paper, that is easy to follow. The introduction is concise but complete. The methodology is clear and sufficient to replicate. The results are presented in a serious of clear figures and tables. The discussion picks up on the most important points and gives the number of pointers for future work.

There are only a few minor points that I suggest the authors to think about and/or change.

  • I would like to see a discussion about why there is a 2.5dB drop post-surgery in the contralateral ear. This is surprising in itself as is the drop in speech perception.
  • The authors might underestimate the effect of a small drop in audibility on speech recognition. In the case of speech in noise tests (which were obviously not done here), a 2.5 dB drop in HL can easily make a 10% difference in speech recognition when at the steepest point in the psychometric function. For the clean speech tests done here, it is assumed that patients are in the saturated part of the individual psychometric functions, but arguably, for hearing impaired listeners that only get a middle percentage, arguably every ‘clean’ speech test is a ‘speech in noise’ test.
  • I suppose this is also at least part of the reason why in table 3 there is a clear increase in ‘Difference’ (between pre and post op) recognition values for people with less hearing. A full answer could only be given if all psychometric functions were measured, which is of course impossible.
  • line 234-235: check. This is a draft. Please provide evidence of ethical approvals.
  • Line 272: this is a corker: for a p-value of 0.05, 2.5% significant left and 2.5% significant rights are expected (for a non-directional hypothesis). Please half your numbers. (doesn’t make a huge difference to the argument though, I think you overestimate the importance of ‘significance’ when measuring a large number of individual noisy psychophysical results with hearing impaired. Much more important would be to look at the effect size). Therefore, I have my doubts about the ‘importance’ (line 269) really.
  • The matlab code is not necessary. It is a simple algorithm that can be described in functionality in a short paragraph.
